# Diagnosis of Acute Myocarditis Using Texture-Based Cardiac Magnetic Resonance, with CINE Imaging as a Novel Tissue Characterization Technique

**DOI:** 10.3390/diagnostics12123187

**Published:** 2022-12-16

**Authors:** Evin I. Papalini, Christian L. Polte, Emanuele Bobbio, Kerstin M. Lagerstrand

**Affiliations:** 1Department of Medical Physics and Biomedical Engineering, Sahlgrenska University Hospital, 41345 Gothenburg, Sweden; 2Institute of Clinical Sciences, Sahlgrenska Academy, University of Gothenburg, 41390 Gothenburg, Sweden; 3Department of Clinical Physiology, Sahlgrenska University Hospital, 41345 Gothenburg, Sweden; 4Department of Radiology, Sahlgrenska University Hospital, 41345 Gothenburg, Sweden; 5Institute of Medicine, Sahlgrenska Academy, University of Gothenburg, 41390 Gothenburg, Sweden; 6Department of Cardiology, Sahlgrenska University Hospital, 41345 Gothenburg, Sweden

**Keywords:** cardiac magnetic resonance, acute myocarditis, myocarditis, texture analysis, texture-based diagnosis, tissue characterization, contrast-free imaging, CINE

## Abstract

Cardiac magnetic resonance (CMR) has emerged as a useful tool in the diagnostic work-up of patients with clinically suspected acute myocarditis (AM), yet the diagnosis remains challenging. The purpose of this proof-of-concept study was to evaluate if data-driven texture analysis has the feasibility to automatically distinguish between patients with and without CMR-verified AM using T2-weighted, late gadolinium enhancement, and CINE imaging. In particular, the present study investigated if functional CINE imaging could be used as a novel tissue characterization technique. Twenty patients with clinically suspected AM, separated into CMR-verified (*n* = 10) and non CMR-verified (*n* = 10) AM according to the Lake Louise criteria, were retrospectively included. Texture features were extracted from the images, compared on a group level, and correlated to the diagnostic outcome (CMR-verified versus non CMR-verified AM). Several features showed good to excellent reproducibility with very large differences between the groups, and moderate to strong correlation with the diagnostic outcome, suggesting that CMR texture analysis is a promising diagnostic tool for patients with clinically suspected AM. Furthermore, findings indicate that CINE imaging, which is currently used for the evaluation of cardiac function, might be a useful non-contrast-based technique for tissue characterization in patients with clinically suspected AM.

## 1. Introduction

Myocarditis is an inflammatory disease of the myocardium with a heterogenous clinical presentation and large spectrum of underlying etiologies [1]. Consequently, the diagnosis of myocarditis remains challenging and the search for further diagnostic markers continues.

Cardiac magnetic resonance (CMR) has emerged as a useful non-invasive imaging tool in the diagnostic work-up of patients with clinically suspected acute myocarditis (AM), where late gadolinium enhancement (LGE) and T2-weighted images are conventionally used to target the myocardial inflammatory process, including the presence of edema and necrosis/fibrosis [2]. While such CMR images are typically assessed in a qualitative way, CMR can also be deployed to collect quantitative information about the myocardial tissue properties. Parametric T1 and T2 mapping for example, allows objective characterization of the regional distribution of AM in terms of the T1 and T2 relaxation times. However, their use in clinical routine has been limited because of the dependence on the chosen scanner and scan protocol, as well as on the patient’s heart rate with a resulting overlap of T1 and T2 relaxation times between healthy and diseased tissue [3,4,5,6]. Thus, other imaging techniques that offer objective evaluation of patients with clinically suspected AM are desirable.

Texture analysis is a novel post-processing technique that attempts to objectively characterize image regions based on their texture content. In this sense, the texture content refers to variations in the intensity values, or gray levels in the image. For example, tissue qualities described by terms such as patchy, focal, rough, or smooth can be mathematically transformed into texture features that characterizes the specific appearance of the tissue quantitatively [7]. The full potential of the technique is not yet adopted as it has the possibility to add new diagnostic information for the detection of pathology that is imperceptible to the human eye, by, for example, exploiting inter-pixel relationships in magnetic resonance images [8]. Its usefulness to distinguish myocardial infarction from AM based on LGE images [9] and to detect acute infarct-like myocarditis on parametric T1 and T2 maps [10] has previously been demonstrated. However, no previous study has evaluated if texture analysis has the feasibility to distinguish between patients with and without AM in a cohort with suspected AM.

In clinical practice, CINE imaging is used to provide morphological and functional information of the heart, focusing on the detection of ventricular dysfunction and pericardial effusion [2]. However, CINE images display a T2/T1-weighted contrast [11] and, as such, should be able to capture active inflammatory processes and the development of fibrosis in the myocardium. In contrast to T2-weighted imaging, the CINE technique is known to be more robust and offers images of high fidelity. Furthermore, this non-contrast-based technique is an attractive alternative to the contrast-based LGE technique, especially in patients with severe renal impairment, in which the administration of a gadolinium contrast agent should be avoided [12]. So far, CINE imaging has not been exploited as a diagnostic tool for tissue characterization, mainly due to its limited visibility for the human eye. However, texture analysis may overcome this obstacle and reveal possible pathological patterns in the CINE image not clearly perceptible to the naked eye.

Accordingly, the aim of this proof-of-concept study was to evaluate the feasibility of texture analysis to distinguish between patients with and without CMR-verified AM using conventional T2-weighted and LGE imaging, as well as CINE imaging. In particular, the present study investigated if functional CINE imaging could be used as a novel tissue characterization technique.

## 2. Materials and Methods

### 2.1. Study Population

Fifty-five patients who underwent a conventional CMR study between February 2013 and October 2017 due to a clinically suspected AM were retrospectively identified from our clinical imaging database. Based on the guidelines of the European Society of Cardiology Working Group on Myocardial and Pericardial Disease, patients with clinically suspected myocarditis were included if ≥1 clinical presentation and ≥1 diagnostic criteria (excluding CMR results) were fulfilled [13,14]. Medical records were scrutinized and a total of thirty-five patients were excluded since their clinical presentation was not consistent with the criteria proposed by the guidelines of the European Society of Cardiology Working Group on Myocardial and Pericardial Disease (*n* = 21) [13,14], or due to incomplete data (*n* = 9) as well as suboptimal image quality (*n* = 5). In the end, we identified ten appropriate patients with CMR-verified AM (*n* = 10) and ten patients with non CMR-verified AM (*n* = 10) based on T2-weigted and LGE imaging according to the original Lake Louise Criteria (LLC) [13,14], which were included in the present feasibility study. The median time from the clinical suspicion to CMR was 8.5 (IQR = 7.3) days for the CMR-verified AM group and 10 (IQR = 8.5) days for the non CMR-verified AM group (*p*-value = 0.65). Basic patient characteristics are summarized in Table 1.

### 2.2. Cardiac MR

CMR imaging was performed on a 1.5T scanner (Philips Medical Systems, Best, The Netherlands) using a five-channel cardiac phased array coil. After standardized patient-specific planning, a series of T2-weighted, LGE, and CINE images were acquired in long and short axis views using retrospective electrocardiographic gating during a gentle expiratory breath hold. T2-weighted images with black blood Short Inversion Time Recovery fat suppression (T2 BB STIR; TR/TE = 1714/70 ms; FOV = 350 × 350 mm^2^; acquisition matrix = 232 × 181; slice thickness = 8 mm; reconstructed in-plane resolution = 0.66 × 0.52 mm^2^) were acquired. LGE images were acquired ~10 min after administration of gadolinium agents (TR/TE = 6.1/3 ms; FOV = 320 × 320 mm^2^; acquisition matrix = 196 × 187; slice thickness = 8 mm; reconstructed in-plane resolution = 0.61 × 0.58 mm^2^). CINE imaging was performed using balanced steady-state-free-precession imaging sequences (TR/TE = 2.8/1.4 ms; FOV = 320 × 320 mm^2^; acquisition matrix = 196 × 168; slice thickness = 8 mm; reconstructed in-plane resolution = 0.61 × 0.48 mm^2^; time frames per cardiac cycle = 40).

CMR analysis, including determination of ventricular volumes and function, and presence of T2 and LGE findings, was performed by a senior imaging expert (C.L.P.). The CMR diagnosis of AM was based on the original LLC using T2 weighted and LGE images [13,14].

### 2.3. Image Segmentation and Texture Analysis

Image segmentation and texture analysis were performed on all short axis CMR images using a freely available software package (MaZda version 4.6; Institute of Electronics, Technical University of Lodz, Lodz, Poland) [15,16]. Three slices in the mid-ventricular area of the heart were selected and analyzed in end-systole, using equivalent slice positions for all imaging techniques. For patients with LGE, slices including apparent LGE were used. All selected images were exported for further analysis as single DICOM (Digital Imaging and Communications in Medicine) images, resulting in 90 images in each group. Subsequently, the left ventricular myocardium was delineated by one reader (E.I.P., with supervision of C.L.P.) in all images (Figure 1), excluding the trabeculated layer and epicardial border to avoid partial volume effects. Images with a non-distinguishable epicardial border were omitted, resulting in 63 images in each group.

Normalization of gray-level values was performed within the texture analysis software by rescaling the histogram data to fit within μ ± 3σ (μ = gray-level mean, σ = gray-level standard deviation), to correct for intra- and inter-scanner intensity variations [17]. Texture features based on image histogram, absolute gradient, run-length matrix, co-occurrence matrix, auto-regressive model, and wavelets (Table 2) were then computed and exported for further analyses. For each imaging technique, 275 texture features were extracted.

To determine the intra-observer reproducibility, the segmentation of the region of interest was repeated once in a subset of ten randomly selected patients, evenly distributed between both study groups. Texture features that were highly reproducible were then compared between groups to study their ability to depict AM.

### 2.4. Statistical Analysis

Statistical calculations were performed with MATLAB (Version 9.10.0, The MathWorks Inc., Natick, Massachusetts, USA). All continuous data are presented as means and standard deviations, unless otherwise stated. The reproducibility was assessed by computing intraclass correlation coefficients (ICC) and their 95% confidence intervals using single measurements, absolute-agreement, and a 2-way mixed-effects model. ICC-values between 0.75 and 1.0 were considered highly reproducible, with a “good” to “excellent” level of agreement [18]. Texture features were correlated to the diagnostic outcome using the Pearson’s correlation coefficient (*R*), where absolute values ≥ 0.4 were regarded as moderate, ≥0.6 as strong, and ≥0.8 as very strong correlation [19]. Testing for group differences was performed using the non-parametric Wilcoxon rank-sum test with the false discovery rate correction to counteract the multiple comparisons. A *p*-value of less than 0.05 was regarded statistically significant.

## 3. Results

### 3.1. Reproducibility Measurements

For all imaging techniques, most of the extracted texture features were found to be highly reproducible. The number of reproducible features was slightly higher for the CINE technique in comparison to the T2-weighted and LGE techniques (248 vs. 237 and 206).

### 3.2. Group Comparisons and Correlation Analysis

As illustrated in Table 3 and Figure 2, very large separations in feature distributions between patients with CMR-verified AM and non CMR-verified AM were found for most texture features. The full list of the texture features with corresponding statistical data is displayed as Appendix A. Furthermore, the CINE technique was found to present the highest number of robust and significant features and the T2-weighted technique the fewest number (*n* = 11, 118, and 161 for T2-weighted, LGE, and CINE). Many of the robust and significant features displayed moderate to strong correlation with AM (*n* = 11, 83, and 136 for T2-weighted, LGE, and CINE) and as shown in Table 3, most of these features were from the same texture category (co-occurrence matrix).

## 4. Discussion

The present proof-of-concept study demonstrates the feasibility of texture analysis to automatically distinguish between patients with and without CMR-verified AM using T2-weighted, LGE, and CINE imaging. In particular, the findings suggest that non-contrast-based CINE imaging can be used to detect myocardial changes associated with AM, even when imperceptible to the human eye. Hence, CINE imaging seems to be a promising tissue characterization technique for the evaluation of AM and, as such, it might be an alternative contrast-free approach for patients with severe renal impairment.

Non-contrast CINE imaging is the gold standard for the assessment of biventricular volumes and function [20] and is included in each CMR protocol in clinical routine. Due to the intrinsic T1- and T2-weighted contrast of the imaging technique, it may also be used for the characterization of the inflammatory process in the myocardium due to AM, as confirmed by present findings. Similar to the conventional imaging techniques, i.e., T2-weighted and LGE imaging, CINE imaging has been shown to offer a promising diagnostic tool for the evaluation of AM. The CINE technique even displayed a larger number of robust features with stronger correlations to the diagnostic outcome than the other imaging techniques, probably attributed to the higher quality of the images. As such, inclusion of CINE imaging in the CMR protocol for tissue characterization might improve the diagnostic work-up of patients with clinically suspected AM. Further, due to the novel image contrast and the possibility of measuring changes in the myocardium dynamically over the whole cardiac cycle, CINE imaging may highlight new phenotypes of AM and contribute to the understanding of the disease. However, future work is needed to confirm this hypothesis.

As recommended by the original LLC, both T2-weighted and LGE imaging are included in the CMR protocol to visualize inflammatory edema and necrosis/fibrosis in the myocardium [2]. Present findings show that these imaging techniques offer texture features that have potential to depict AM in a reproducible manner. However, the T2-weighted technique presented less reproducible features. This was probably attributed to the higher artefactual signal level in these images [21,22]. Still, such data may improve the classification of AM, as it may add novel information that reflects ongoing inflammation.

Texture-based analysis has the advantage of being able to extract quantitative markers from qualitative images, and, as such, has the feasibility to be used in longitudinal follow-up studies of patients with clinically suspected AM. Further, this data-driven analysis technique enables automatic extraction of image-based features and, thereby, offers more time efficient and robust evaluations of patients with clinically suspected AM. Since such analyses will not rely on the ability of an observer, subtle changes in the myocardial tissue due to an AM may not go undetected. A large variation in the diagnostic performance of the original LLC has been reported by several studies (AUC = 57–90%), probably due to the fact that it depends on a qualitative analysis of signal intensities on T2-weighted, early gadolinium enhancement, and LGE images, and hence, may miss cases with subtle myocardial changes [2]. Introducing automatic texture-based analysis in the evaluation might further improve the diagnostic performance of CMR in patients with clinically suspected AM. As shown here by the present findings, all included imaging techniques offer several texture features with very large differences between the groups and strong associations to the diagnostic outcome, displaying their feasibility in detecting AM on an individual level. Hence, the present approach shows promise in building a classification model for the diagnosis of AM.

A strength of the present study design was the inclusion of a relevant reference cohort to address the diagnostic clinical reality. As such, the study did not use healthy controls as reference but included only patients with clinically suspected AM (CMR-verified AM versus non CMR-verified AM). However, the retrospective nature of the study and its small number of included patients might have introduced a selection bias. Even though the present cohort displayed very large differences between groups and was sufficiently large for a feasibility study, the true diagnostic potential of CINE imaging and texture-based CMR imaging as biomarkers for the diagnosis and phenotyping of AM needs to be evaluated in larger prospective cohort studies. Accordingly, the results of this proof-of-concept study should be interpreted with caution.

## 5. Conclusions

The present feasibility study shows that texture analysis is a promising diagnostic tool for patients with clinically suspected AM using T2-weighted, LGE, and CINE imaging. Furthermore, the findings suggest that CINE imaging, which is used in clinical routine for the evaluation of cardiac function, might be a useful novel non-contrast-based technique for tissue characterization in patients with clinically suspected AM. However, future studies are needed to substantiate our findings.

## Figures and Tables

**Figure 1 diagnostics-12-03187-f001:**
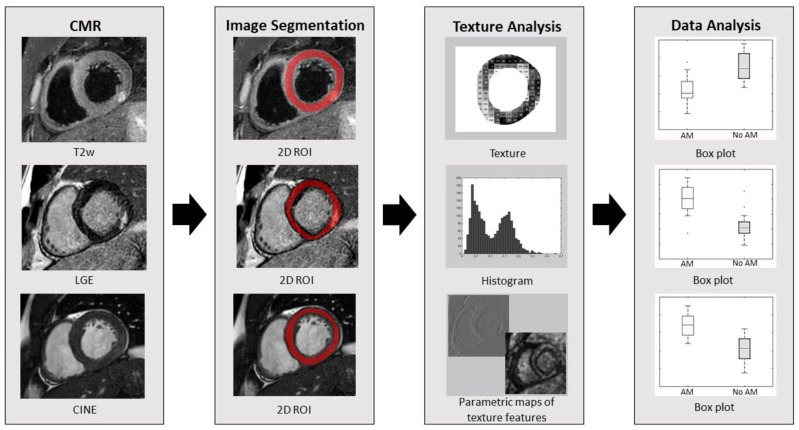
Schematic illustration of the workflow of the study, including the texture analysis. In a cohort of patients with suspected acute myocarditis (AM), texture features were extracted from conventional cardiac magnetic resonance (CMR) images, including T2-weighted (T2w), late gadolinium enhancement (LGE) and CINE images, using predefined myocardial region of interests (2D ROI). Extracted features were compared between patients with and without CMR-verified AM.

**Figure 2 diagnostics-12-03187-f002:**
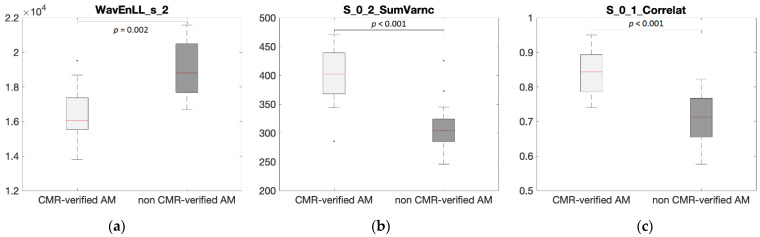
Box plots illustrating the large difference in myocardial tissue textures in (**a**) T2-weighted, (**b**) late gadolinium enhancement, and (**c**) CINE images between patients with CMR-verified AM and non CMR-verified AM. In specific, WavEnLL_s_2, S_0_2_SumVarnc, and S_0_1_Correlat that presented the highest correlation with AM (Table 3) are displayed. The median is presented by the centerline with the upper and lower limits of 25th and 75th percentiles, respectively. Texture features are dimensionless.

**Table 1 diagnostics-12-03187-t001:** Basic patient characteristics.

Characteristic	Clinically Suspected AM (*n* = 20)	CMR-Verified AM (*n* = 10)	Non CMR-Verified AM (*n* = 10)	*p*-Value
Age (years)	28 (28)	25 (10)	49 (24)	0.001 ^‡^
No. of men	18	9	9	1
Height (cm)	180 (8)	177 (19)	181 (4)	0.44
Weight (kg)	81 (22)	78 (34)	86 (18)	0.33
Symptom ^†^				
Fever	5	4	1	-
Fatigue	3	1	2	-
Chest pain	19	10	9	-
Dyspnea	3	1	2	-
Recent infection	14	10	4	-
No. of diagnostic criteria for clinically suspected AM ^¶^	3.0 (1.8)	3.5 (1.0)	3.0 (1.0)	0.29
Pathologic ECG finding ^†^				
Sinus rhythm	20	10	10	-
AV conduction abnormality	1	0	1	-
ST-segment elevation	7	4	3	-
ST-segment depression	3	1	2	-
T-wave inversion	6	3	3	-
Pathologic blood result ^†^				
TNT/TNI	30 (538)	606 (1407) ^§^	24 (40)	0.001 ^‡^
NT-proBNP	278 (487)	424 (314) ^§^	17 (140) ^§^	0.003 ^‡^
CRP	16 (35)	34 (60) ^§^	7 (16)	0.07
Cardiovascular risk factor ^†^				
Hypertension	2	1	1	-
Hyperlipidaemia	0	0	0	-
Diabetes	0	0	0	-
Smoker	0	0	0	-
Obesity	1	0	1	-
CMR findings				
LV iEDV (mL)	91 (19)	91 (13)	86 (37)	0.10
LV iESV (mL)	37 (13)	38 (10)	35 (23)	0.17
LV iSV (mL)	50 (12)	53 (10)	46 (16)	0.08
LV EF (%)	59 (6)	59 (5)	59 (8)	0.39
T2w findings ^†^	10	10 *	0	-
LGE findings ^†^	10	10	0	-

Unless stated otherwise, data are expressed as medians (interquartile range). AM: acute myocarditis; AV: atrioventricular; CMR: cardiac magnetic resonance; CRP: C-reactive protein; ECG: electrocardiogram; EDV: end-diastolic volume; EF: ejection fraction; ESV: end-systolic volume; T2w: T2-weighted; LGE: late gadolinium enhancement; LV: left ventricle; NT-proBNP: N-terminal pro–B-type natriuretic peptide; SV: stroke volume; TNI: Troponin I; TNT: Troponin T. ^†^ Data indicate number of affected participants. ^‡^ Indicates statistical significance (*p* < 0.05) between patients with and without CMR-verified AM. ^¶^ CMR findings were excluded from the calculation [14]. ^§^ Missing data for CMR-verified AM: TNT, *n* = 3; NT-pro-BNP, *n* = 4; CRP, *n* = 1 and non CMR-verified AM: NT-pro-BNP, *n* = 4. * Median T2 signal intensity ratio between pathologic and control tissue = 2.3.

**Table 2 diagnostics-12-03187-t002:** Extracted texture categories with corresponding texture features.

Texture Category	Texture Feature	Number of Features
Histogram	Mean, variance, skewness, kurtosis, percentiles (1%, 10%, 50%, 90%, 99%).	9
Absolute gradient (4 bits/pixel)	Gradient mean, variance, skewness, kurtosis, non-zeros.	5
Run-length matrix (computed for four angles (vertical, horizontal, 45°, and 135°); 6 bits/pixel)	Run-length non-uniformity, gray-level non-uniformity, long run emphasis, short run emphasis, fraction of image in runs.	20
Co-occurrence matrix (computed for four directions ((x,0), (0,x), (x,x), (x,−x)) at five interpixel distances (x = 1–5); 6 bits/pixel)	Angular second moment, contrast, correlation, entropy, sum entropy, sum of squares, sum average, sum variance, inverse different moment, difference entropy, difference variance.	220
Autoregressive model	Teta 1 to 4, sigma.	5
Wavelet transform (calculated for four subsampling factors (*n* = 1–4); 8 bits/pixel)	Energy of wavelet coefficients in low-frequency sub-bands, horizontal high-frequency sub-bands, vertical high-frequency sub-bands, and diagonal high-frequency sub-bands.	16

**Table 3 diagnostics-12-03187-t003:** Calculated means and standard deviations of ten robust and significant texture features with the highest correlation with AM, extracted from the T2-weighted, LGE, and CINE images. Corresponding ICC-values for reproducibility, *p*-values for comparisons between groups, and *R*-values for the association with diagnostic outcome are also presented.

CMR	Texture Feature	CMR-Verified AM	Non CMR-Verified AM	ICC	*p*-Value	*R*
T2w	WavEnLL_s_2	16,400 ± 1360	19,100 ± 1600	0.85	0.002	−0.67
	WavEnLL_s_3	12,500 ± 1690	16,900 ± 3570	0.88	0.008	−0.62
	S_3__3_SumOfSqs	113 ± 8	103 ± 5	0.88	0.012	0.59
	S_2__2_SumOfSqs	111 ± 6	104 ± 4	0.81	0.012	0.59
	WavEnLL_s_4	8050 ± 1920	12,200 ± 3600	0.94	0.012	−0.58
	S_1_1_SumAverg	64.7 ± 0.8	63.8 ± 0.6	0.77	0.012	0.56
	S_2_2_SumAverg	65.2 ± 1.2	63.8 ± 0.9	0.89	0.019	0.54
	S_4__4_SumOfSqs	113 ± 9	102 ± 10	0.81	0.019	0.50
	Skewness	0.37 ± 0.38	−0.01 ± 0.3	0.89	0.031	0.49
	S_0_2_SumOfSqs	110 ± 5	106 ± 4	0.81	0.021	0.48
LGE	S_0_2_SumVarnc	401 ± 46	306 ± 42	0.98	<0.001	0.73
	S_0_4_SumVarnc	376 ± 73	233 ± 65	0.96	<0.001	0.72
	S_0_5_SumVarnc	372 ± 82	228 ± 66	0.97	<0.001	0.69
	S_1__1_SumVarnc	411 ± 28	258 ± 29	0.95	<0.001	0.69
	S_0_1_SumVarnc	424 ± 24	379 ± 23	0.98	<0.001	0.68
	S_2_0_SumAverg	65.6 ± 0.9	63.6 ± 1.3	0.78	<0.001	0.68
	S_0_2_SumAverg	65.9 ± 1.1	63.6 ± 1.4	0.77	<0.001	0.68
	S_0_2_SumOfSqs	114 ± 6	102 ± 7	0.89	<0.001	0.66
	WavEnLL_s_2	17,700 ± 1540	21,200 ± 2320	0.91	<0.001	−0.66
	S_0_4_Correlat	0.57 ± 0.21	0.18 ± 0.23	0.98	<0.001	0.66
CINE	S_0_1_Correlat	0.84 ± 0.06	0.72 ± 0.07	0.98	<0.001	0.70
	S_0_2_Correlat	0.61 ± 0.15	0.32 ± 0.15	0.99	<0.001	0.70
	S_0_2_SumVarnc	359 ± 50	274 ± 39	0.98	<0.001	0.69
	S_0_1_Contrast	34.0 ± 13.2	59.9 ± 13.9	0.97	<0.001	−0.68
	S_0_2_Contrast	85.3 ± 31.8	141 ± 28	0.97	<0.001	−0.68
	S_1__1_Correlat	0.73 ± 0.10	0.54 ± 0.11	0.98	<0.001	0.67
	S_1__1_Contrast	57.8 ± 20.3	94.9 ± 21.5	0.98	<0.001	−0.66
	S_0_1_SumVarnc	405 ± 25	362 ± 25	0.89	<0.001	0.66
	S_0_3_Correlat	0.51 ± 0.19	0.20 ± 0.17	<1.00	<0.001	0.66
	Sigma	0.35 ± 0.08	0.48 ± 0.08	0.98	<0.001	−0.65

AM: acute myocarditis; CMR: cardiac magnetic resonance; ICC: intraclass correlation coefficient; LGE: late gadolinium enhancement; R: Pearson’s correlation coefficients; T2w: T2-weighted.

## Data Availability

The analyzed data sets generated during this study are available from the corresponding author on reasonable request.

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
