# Peer review of "Diagnosis of Acute Myocarditis Using Texture-Based Cardiac Magnetic Resonance, with CINE Imaging as a Novel Tissue Characterization Technique"

_diagnostics, 2022, doi:10.3390/diagnostics12123187_

Round 1

Reviewer 1 Report (Previous Reviewer 1)

I thank the Authors for reviewing their manuscript, especially for clarifying points relevant to the understanding of the manuscript. I do, however, still have some concerns.

Methods

Although I appreciate this study is intended to be a feasibility study, the sample size is really very small. The robustness of the findings would be increased by confirming these findings in an equivalent number of clinically suspected acute myocarditis patients from an external validation cohort.

As there is no endomyocardial biopsy proven case, it would be useful to know how many clinical and diagnostic criteria were met in patients from the two groups (this could be added as mean value in the overall cohort and in both CMR groups, in Table 1), to strengthen the diagnosis of clinically suspected acute myocarditis, especially in the non-CMR verified group.  

The Authors state that old Lake Louise criteria (LLC) were used to define CMR-based acute myocarditis.  By definition, two out of three criteria (edema, early and late gadolinium enhancement) should be met, according to old LLC; however, there is no mention to early gadolinium enhancement (EGE) sequences, which were integral part of the assessment of suspected acute myocardial inflammation according to the old LLC used by the Authors. Please explain why these data are missing.

It would also be interesting to know if anyone of the non-CMR verified group had positive EGE sequences.

Timing between clinical diagnosis of acute myocarditis and CMR should be specified for both groups, as some of the non-CMR verified patients may have been scanned either too early or too late to detect edema, for instance.

The Authors should briefly explain how texture analysis works, for non-expert readers.

CMR was not assessed blinded to patient’s diagnosis, as all patients were known to have clinically suspected acute myocarditis; please amend the text.

Tables and figures

Please provide characteristics of the entire population of clinically suspected myocarditis (n=20) in Table 1 by adding an extra column.

Please add p-values in Figure 2.

Author Response

Reviewer 2 Report (Previous Reviewer 3)

I read the revised version of the article entitled “Doagnosos of acute myocarditis using texture-based magnetic resonance, with CINE imaging as a novel tissue characterization technique” and I think that the corretions made by the authors are appropriate and have improved the content of the article. I confirm that the authors correctly conducted the project study and used the appropriate materials and methods. In addition, the results reflect the purpose of the study and the reading of the article can be useful for the readers of the journal. Therefore, I think that this article is suitable for publication in its current revised version.

Author Response

Thank you for your positive response!

This manuscript is a resubmission of an earlier submission. The following is a list of the peer review reports and author responses from that submission.

Round 1

Reviewer 1 Report

The Authors aimed at evaluating the accuracy of texture analysis to diagnose acute myocarditis using conventional CMR tissue characterization sequences as well as cine sequences. They found that texture analysis can be used for automated diagnosis of acute myocarditis using conventional T2-weighted, late gadolinium enhancement, and cine imaging. 

- As far as I understand, the 2013 ESC consensus was used to define clinically suspected myocarditis, patients could then have CMR features of myocarditis or not. This should be however more clearly specified. 

- Why were only 10 myocarditis patients included in the final analysis? What happened to the remaining 45 patients?

- It is not clear who the controls are: are they clinically suspected myocarditis  patients with negative CMR (it would not be correct to consider these as controls, as a patient can have myocarditis with a negative CMR)? Or are they healthy controls? 

- The sample size is far too small to draw any conclusion; Authors should consider increasing sample size or should provide power analysis to support such a small sample size

- Rather than against controls, texture analysis findings should have been compared to standard CMR diagnostic criteria for myocarditis (even the old Lake Louise Criteria or, even better, the updated Lake Louise Criteria)

Reviewer 2 Report

Interesting work, raises an important clinical problem, which is the diagnosis of myocarditis. The authors proposed texture analysis, a novel post-processing technique, that can objectively characterize tissue changes on T2-weighted, LGE and CINE images. The study group consisted of 10 patients with myocarditis and 10 people in the control group. A small number of respondents is the weakest point of work, thus results should be treated with caution. The authors excluded 35 patients from a group of 55. Could the authors provide us the information, how many patients were excluded for each of following reasons: incomplete data, insufficient image quality, pre-existing cardiac disease and significant coronary artery disease.

Myocarditis is often complicated by arrhythmias and tachycardia and obtaining very good quality images in this group of patients can be a big challenge, and often impossible to achieve. Therefore, further research is needed to demonstrate the effectiveness and advantage  of texture analysis over LLC in a group of patients who, for various reasons, did not obtain the highest quality images.

Same  minor revision of the manuscript should be done:

-              Please add information what kind of fat saturation was used in T2-weighted images

-              Please correct the units: FOV  in mm, acquisition matrix in pixel, in-plane resolution in mm x mm

-              Patients preselection should be added to study limitation.

Reviewer 3 Report

I read the articje entitled “Diagnosis of acute myocarditis using texture-based cardiac magnetic resonance, with CINE imaging as novel tissue characterization technique” and I think that  this argument is clinically important in the diagnostic work-up of the myocarditis. This retrospective study compares two different  techniques of cardiac magnetic resonance (CMR) such as LGE and CINE and found that the CINE is a good technique with the favorable aspect that it is contrast-free. I think that this article is correctly write in terms of originality and methods and the references are appropriate. I have no questions. Therefore, I think that this article is suitable for publication in its current version
